# Cost-Effectiveness of Treatment Optimisation with Biomarkers for Immunotherapy in Solid Tumours: A Systematic Review

**DOI:** 10.3390/cancers16050995

**Published:** 2024-02-29

**Authors:** Sara Mucherino, Valentina Lorenzoni, Isotta Triulzi, Marzia Del Re, Valentina Orlando, Annalisa Capuano, Romano Danesi, Giuseppe Turchetti, Enrica Menditto

**Affiliations:** 1CIRFF—Centre of Pharmacoeconomics and Drug Utilization Research, Department of Pharmacy, University of Naples Federico II, via D Montesano 49, 80131 Naples, Italy; sara.mucherino@unina.it (S.M.); valentina.orlando@unina.it (V.O.); 2Institute of Management, Scuola Superiore Sant’Anna, 56127 Pisa, Italy; valentina.lorenzoni@santannapisa.it (V.L.); isotta.triulzi@santannapisa.it (I.T.); giuseppe.turchetti@santannapisa.it (G.T.); 3Unit of Clinical Pharmacology and Pharmacogenetics, Department of Clinical and Experimental Medicine, University of Pisa, 56126 Pisa, Italy; marzia.delre@ao-pisa.toscana.it (M.D.R.); romano.danesi@unipi.it (R.D.); 4Section of Pharmacology ‘L. Donatelli’, Department of Experimental Medicine, University of Campania ‘L. Vanvitelli’, Via Costantinopoli 16, 80138 Naples, Italy; annalisa.capuano@unicampania.it

**Keywords:** immunotherapy, biomarkers, cost effectiveness, economic evaluation, quality of life

## Abstract

**Simple Summary:**

Researchers aim to assess the economic impact of testing predictive biomarkers for immunotherapy in solid tumour treatment using immune checkpoint inhibitors (ICIs). Despite recent advancements, concerns persist regarding the cost-effectiveness and budgetary implications of ICIs. This study systematically reviewed the economic evaluations of biomarker testing from various databases. The team assessed studies from June 2010 to February 2022, evaluating their quality and synthesising findings by tumour type. Understanding the economic implications of these tests could help drive future research, optimise treatment strategies, potentially influencing health care decisions and resource allocation in solid tumour therapy, impacting how we approach and fund immunotherapy for better patient outcomes.

**Abstract:**

This study investigated the health economic evaluations of predictive biomarker testing in solid tumours treated with immune checkpoint inhibitors (ICIs). Searching PubMed, EMBASE, and Web of Science from June 2010 to February 2022, 58 relevant articles were reviewed out of the 730 screened. The focus was predominantly on non-small cell lung cancer (NSCLC) (65%) and other solid tumours (40%). Among the NSCLC studies, 21 out of 35 demonstrated cost-effectiveness, notably for pembrolizumab as first-line treatment when preceded by PD-L1 assessment, cost-effective at a threshold of $100,000/QALY compared to the standard of care. However, for bladder, cervical, and triple-negative breast cancers (TNBCs), no economic evaluations met the affordability threshold of $100,000/QALY. Overall, the review highlights a certain degree of uncertainty about the cost-effectiveness of ICI. In particular, we found PD-L1 expression associated with ICI treatment to be a cost-effective strategy, particularly in NSCLC, urothelial, and renal cell carcinoma. The findings suggest the potential value of predictive biomarker testing, specifically with pembrolizumab in NSCLC, while indicating challenges in achieving cost-effectiveness for certain other solid tumours.

## 1. Introduction

The advent of immunotherapy has significantly changed the therapeutic scenario of cancer patients. Immunotherapy drugs, so-called immune checkpoint inhibitors (ICIs), work by blocking checkpoint proteins from binding to their partners, representing a new weapon for cancer treatment in different settings [1]. Moreover, the association of a biomarker test to identify the best treatment choice has revolutionised patient management for many tumour types, driving towards a patient-tailored approach [2]. Since the FDA approval of ipilimumab (cytotoxic T-lymphocyte-associated antigen-4 (CTLA-4) inhibitor) in 2011, six more ICIs have been approved for cancer therapy: the programmed cell death-1 (PD-1) inhibitors including nivolumab, pembrolizumab, cemiplimab, and the programmed cell death ligand-1 (PD-L1) inhibitors including atezolizumab, durvalumab, and avelumab. These agents have become the standard of care in many solid tumours [1,2,3]. Thus, a predictive biomarker testing approach in this field may represent a virtuous model to invest in for treatment optimisation, with a possible impact also on the patients’ quality of life (QoL). Despite the successes achieved in recent years, there are many concerns about the cost-effectiveness (CE) and budget impact of the next wave of ICIs [3]. This systematic review, as part of a funded Italian National Research Project, aims to provide a snapshot of the current state-of-the-art regarding the cost-effectiveness, cost-utility, or net-monetary benefits of the use of predictive biomarkers in solid tumours treated with ICIs as tools for customising immunotherapy.

## 2. Materials and Methods

A systematic literature review was conducted in accordance with the PRISMA 2020 statement guidelines [4]. The review methodology was prospectively registered a priori in PROSPERO (CRD42020201549) as for the entire study protocol [5].

### 2.1. Data Sources and Searches

Searches of relevant studies were conducted across the following electronic bibliographic databases: Ovid MEDLINE, EMBASE, Web of Science, from June 2010 to February 2022 as per the date of the first ICI approval [6]. Search strategy and syntaxes are detailed in the Appendix A, respectively. References were collected with Reference Manager (Institute for Scientific Information, Berkeley, CA, USA, ver.12). 

### 2.2. Study Selection, Data Extraction, and Quality Assessment

Predefined criteria for selecting studies related to economic evaluation (population, interventions and comparisons, outcomes, study design, [PICOS]) were set and published in the study protocol [5]. Data were processed by four researchers to identify potentially eligible studies and validated by five clinicians/researchers. The screening process is detailed in Figure 1. The quality of evidence was evaluated by the GRADE approach [7].

### 2.3. Data Synthesis

Country, treatment line, comparators, biomarker test, willingness-to-pay (WTP) threshold, cost-effectiveness ratio, and affordability according to the WTP set were extracted from each study. Information was collected separately for non-small cell lung cancer (NSCLC) and other solid tumour diagnoses. A time-trend analysis was performed. After converting differences in costs to the 2021 US dollar, adjusting for both inflation and purchasing power parities [8] resources, adjusted differences in costs and QALYs were plotted on a CE plane, evaluated with a reference WTP threshold of $100,000/QALY. 

### 2.4. Role of the Funding Source

This research was funded by the Ministero dell’Istruzione, dell’Università e della Ricerca (MIUR) within the framework of the PRIN Project 2017, grant number 2017NR7W5K. This funding source had no role in the design, execution, analyses, interpretation of the data, and results of the present study.

## 3. Results

### 3.1. Study Identification

A total of 730 articles were identified through the systematic literature search across different bibliographic database queries. As reported in Figure 1, 295 (40.4%) duplicates were removed and 435 abstracts (59.6%) were screened, resulting in 76 studies potentially evaluable. After excluding six papers for which the full-text was not retrieved, a total of 70 full-text articles were examined. Among these papers, 10 systematic reviews and one paper based on a budget impact analysis model were excluded, for a total of 58 studies that met the inclusion criteria.

### 3.2. Study Quality

Quality of evidence and risk of biases were assessed based on the GRADE scale for each study. Overall, 60% of economic evaluation (*n* = 35) was based on a Markov model and 26% on the partitioned survival model (*n* = 15). All of the methodological aspects of individual studies are shown in Appendix A. Overall, most economic evaluations (62.1%, *n* = 36 studies) presented a moderate/high certainty of evidence and strength of recommendations. Next, 25.9% (*n* = 15) of the studies achieved a moderate certainty rating, 6.9% (*n* = 4) a high, and 5.2% (*n* = 3) a moderate/low certainty score (Appendix A).

### 3.3. Study Characteristics

The geographical distribution of papers reporting an economic evaluation of an ICI associated to a biomarker test showed a marked majority of studies conducted in the US (56%; *n* = 33) and China (27%; *n* = 16). Figure 2 shows the worldwide distribution of studies included. 

In the overall analysis, 97% of the included studies (*n* = 56) assessed the cost-effectiveness ratio of an ICI with prior PD-L1 testing, whereas only 3% (*n* = 2) considered prior PD-1 testing. Among the overall 58 included studies, 60% (*n* = 35) included patients affected by NSCLC (Table 1) and 40% other solid tumours including 25.9% (*n* = 6) head and neck cancer squamous cell carcinoma (HNSCC), 15.5% (*n* = 9) urothelial, bladder cancer, and renal cell carcinoma (RCC), followed by a few studies on triple-negative breast cancer (5.2%; *n* = 3), melanoma including Merkel cell carcinoma (mMCC) (5.2%; *n* = 3), cervical, and gastric cancers (3.4%; *n* = 2) (Table 2). Following a bibliometric analysis, a huge increase in the number of papers on NSCLC from 2016 to 2021 was observed (Figure 3). This was confirmed by the time-trend estimates of the economic evaluations of ICIs associated with biomarker testing in NSCLC being considerably higher than in other solid tumours.

### 3.4. Non-Small Cell Lung Cancer (NSCLC)

Among the 35 studies on patients with NSCLC, slightly more than half of those studies (63%, *n* = 22) [9,10,11,12,13,14,20,21,22,23,29,30,31,32,37,38,39,42,67,68,69] suggested the cost-effectiveness of ICI treatment with biomarker testing; of these, *n* = 2 with previous PD-1 assessment and *n* = 20 with previous PD-L1 assessment. Particularly, when distinguishing between anti PD-1 and anti PD-L1 treatment and considering the multiple analyses performed as part of the selected studies, there were no differences in the proportion of analyses demonstrating the cost-effectiveness of ICIs (*p*-value = 1), Figure 4. 

In more detail, pembrolizumab (*n* = 17) [9,10,11,13,21,23,24,29,30,31,32,37,38,39,42] was cost-effective with respect to chemotherapy (SoC) when associated with a previous PD-1/PD-L1 assessment, in the first-line treatment of NSCLC [9,10,11,13,21,23,24,29,30,31,32,34,37,38,42]. Moreover, four studies [14,22,39,40] demonstrated the cost-effectiveness of nivolumab with previous PD-L1 expression compared to chemotherapy in both first [22,40] and second [39] line treatment. Furthermore, durvalumab treatment in PD-L1-positive patients [20] was associated with a positive cost-effectiveness profile when compared with the placebo. The same evidence was recorded for atezolizumab from a US perspective [15]. Durvalumab was considered cost-effective with respect to the SoC in the US (WTP of $100,000–$150,000/QALY), but not-cost-effective in China (WTP of $33,210/QALY). Overall, 14 studies [16,17,18,19,25,26,27,28,33,35,36,40,41,43] showed that all ICI types associated with a previous biomarker test were not a cost-effective solution for NSCLC patients in settings where there was a WTP less than $100,000/QALY, also considering overall survival (OS) as most influential factor for the incremental cost-effectiveness ratio. 

Figures show incremental QALY and incremental costs for different immune checkpoint inhibitors (ICI) among NSCLC patients according to NSCLC type (identified by the type of marker) and immune checkpoint inhibitor (identified by marker colour); the size of the marker identifies first-line (bigger) or second-line (smaller) treatment; the area under the WTP threshold indicates the cost-effective studies and is highlighted in green. Studies for which results are available for more than one country are represented more than once.

### 3.5. Other Tumours

#### 3.5.1. Urothelial, Bladder Cancer, and Renal Cell Carcinoma (RCC)

When considering other tumours, statistically significant differences emerged between anti PD-1 and anti PD-L1, with a clear indication of no cost-effectiveness of available evidence for the latter approach in those tumours (Figure 5).

In more detail, five economic evaluations [44,45,46,47,49] were carried out on patients diagnosed with urothelial cancer, demonstrating that avelumab (*n* = 1) [45] and pembrolizumab (*n* = 2) [46,47] were cost-effective in first-line treatment when preceded by PD-L1 testing (with thresholds set in the US and Switzerland) with greater impact on the survival benefit of patients. Conversely, first-line treatment with atezolizumab (*n* = 1) [44] and second-line treatment with pembrolizumab [49] (*n* = 1) were not cost-effective with a threshold of $150,000/QALY in a US setting.

Regarding renal cell carcinoma from a US perspective, nivolumab was cost-effective compared to everolimus [50], and nivolumab with ipilimumab was cost-effective compared to sunitinib in both cases with a PD-L1 test, and a $150,000/QALY threshold [48] generated a gain of 0.34 QALYs and 0.98 QALYs, respectively.

Moreover, the two studies [51,52] conducted on bladder cancer showed that the second-line treatment with pembrolizumab and atezolizumab were not cost-effective compared to chemotherapy in a Canadian, British, and Australian perspective with a WTP less than $100,000/QALY (Figure 5).

#### 3.5.2. Head and Neck Cancer Squamous Cell Carcinoma (HNSCC)

Four out of six studies [53,54,55,56,57,58] revealed that pembrolizumab [53,54,55] and nivolumab [57] were a cost-effective option compared with SoC in the US, China, Switzerland, and Argentina, while two found that nivolumab was not a cost-effective option compared to the SoC in the US [58] and docetaxel in Canada [56].

#### 3.5.3. Triple-Negative Breast Cancer (TNBC)

The cost-effectiveness analysis of atezolizumab plus nab-paclitaxel versus nab-paclitaxel alone provided similar results. Two Markov models and a multi-country partitioned survival model [59,60] indicated that this combination (atezolizumab plus nab-paclitaxel) was not cost-effective compared to nab-paclitaxel alone, both in China and the US. Similarly, another Markov model-based study [61] corroborated these findings across the overall population (ICER $281,448/QALY; WTP of $200,000/QALY) (Figure 5).

#### 3.5.4. Melanoma including Merkel Cell Carcinoma (mMCC)

Three cost-effectiveness analyses were retrieved on ICIs used for the treatment of melanoma and Merkel cell carcinoma (MCC) [62,63,64]. For metastatic MCC, avelumab treatment with previous PD-L1 assessment was cost-effective compared to best supportive care (ICER USD 44,885.06/QALY, WTP USD 53,333.33/QALY) and chemotherapy (ICER USD 42,993.06/QALY) [64] (Figure 5). Regarding patients with advanced melanoma, nivolumab was the most cost-effective treatment option in BRAF wild-type and BRAF mutant patients, as demonstrated by the Markov model developed to estimate the lifetime costs and benefits of nivolumab versus ipilimumab and dacarbazine (for BRAF wild-type) and versus ipilimumab, dabrafenib, and vemurafenib (for BRAF mutant patients) [62]. In patients affected by BRAF wild-type advanced melanoma, the discrete simulation event model of Tarhini et al. [63] found that the most cost-effective treatment sequences initiated with anti-PD-1 + anti-CTLA-4 (nivolumab + ipilimumab and chemotherapy through a mix of dacarbazine, temozolomide, paclitaxel, and carboplatin + paclitaxel), followed by either chemotherapy or anti-PD-1 monotherapy (nivolumab and pembrolizumab, assuming an equal share). This provided substantial quality-adjusted survival gains to patients with BRAF wild-type advanced melanoma.

#### 3.5.5. Cervical and Gastric Cancers

Only one cost-effectiveness analysis for the treatment of patients with persistent, recurrent, or metastatic cervical cancers who had not received systemic chemotherapy and were not amenable to curative treatment was found [65]. Through a partitioned survival model over a 30-year lifetime horizon, Shi et al. [65] indicated that pembrolizumab was not cost-effective versus the placebo. With respect to patients in second-line therapy with metastatic gastric cancer, the most effective strategy was pembrolizumab for high microsatellite instability (MSI-H) patients and ramucirumab/paclitaxel for all other patients, but the Markov model resulted in a high ICER of $1,074,620/QALY. Among the following strategies, pembrolizumab monotherapy and ramucirumab/paclitaxel combination therapy for all patients and pembrolizumab for patients based on MSI status or PD-L1 expression, the only cost-effective one was paclitaxel monotherapy for all patients, with an ICER of $53,705/QALY (WTP USD 100,000/QALY in the US) [66] (Figure 5).

Figures show incremental QALY and incremental costs for different immune checkpoint inhibitors (ICI) for cancer other than NSCLC according to cancer type (identified by the type of marker) and immune checkpoint inhibitor (identified by marker colour); the size of the marker identifies first-line (bigger) or second-line (smaller) treatment; the area under the WTP threshold indicates the cost-effective studies and is highlighted in green. Studies for which results are available for more than one country are represented more than once.

## 4. Discussion

This systematic review examined the cost-effectiveness profile of ICI therapy in solid tumours associated with the biomarker test in oncology. Most of the studies [9,10,11,12,13,14,15,16,17,18,19,20,21,22,23,24,25,26,27,28,29,30,31,32,33,34,35,36,37,38,39,40,41,42,43] included focused on non-small cell lung cancer (NSCLC), followed by HNSCC and genitourinary cancers.

This review highlights that there is still no clear evidence regarding the cost-effectiveness of anti PD-L1 and PD-1 for treating NSCLC, while at present, the available evidence is generally not in favour of the cost-effectiveness of anti PD-L1 in other solid tumours.

Despite there being a significantly fewer number of studies retrieved for other solid tumours, the PD-L1 testing approach was also cost-effective in some settings for urothelial cancer patients first-line treated with avelumab and pembrolizumab but also for renal cell carcinoma treated with nivolumab. Moreover, this approach was also cost-effective in urothelial cancer and renal cell carcinoma.

Consistent with these findings, the European Society for Medical Oncology (ESMO) NSCLC guidelines emphasise the importance of molecular subtyping in guiding therapeutic decision-making, advocating its execution whenever feasible [70,71]. Hence, for patients with advanced NSCLC, ESMO recommends determining PD-L1 expression between others such as EGFR testing, BRAF mutations, and the analysis of ALK, ROS1, and NTRK rearrangements [71]. Generally, there is consensus across international guidelines around the need for PD-L1 testing and other biomarkers in advanced NSCLC. These have approved first-line targeted therapies in Europe [72,73,74]. 

Hence, multiple studies have highlighted the potential of PD-L1 overexpression as a crucial and extensively studied predictive biomarker for assessing the response to PD-L1 antibodies, resulting in improved clinical outcomes [75,76,77,78]. However, the implementation of these novel and more effective therapies has been acknowledged as both essential and cost-prohibitive [79]. To investigate this, the economic evaluations included in this review [9,10,13,21,24,29,30,31,32,34,37,42] confirmed that NSCLC patients with high tumour PD-L1 levels with a proportional score ≥ 50% for first-line therapy with pembrolizumab, exhibited superior response rates to immunotherapy and experienced prolonged survival compared to those who underwent conventional chemotherapy. Building upon these widely recognised clinical findings, this systematic review unequivocally demonstrated that the treatment approach in question was not only clinically effective but also cost-effective. The economic evaluations indicated a favourable cost threshold, falling within the range of $100,000/QALY to $150,000/QALY. Contrary to the majority of pharmacoeconomic assessments, the pivotal result of this review revealed that a distinct subset of studies failed to demonstrate the cost-effectiveness of PD-L1 testing and pembrolizumab for the first-line treatment of non-small cell lung cancer (NSCLC). Crucially, this divergence in outcomes emerged exclusively within a specific setting characterised by a willingness-to-pay (WTP) threshold falling below $100,000/QALY. This singular observation underscores the significance of the WTP threshold as a decisive factor in determining the cost-effectiveness of this therapeutic approach for NSCLC [17,23,26,28,35,36,38,40,41]. Additionally, of paramount importance is that biomarker testing prior to nivolumab treatment is not deemed cost-effective when administered as a second-line treatment [27], in combination with another monoclonal antibody [18], or when the WTP threshold falls below $100,000/QALY [27,43].

Regarding other solid tumours such as in the case of bladder [51,52], cervical [65], and triple-negative breast cancer (TNBC) [59,60,61], no economic evaluation has deemed the costs of the biomarker testing strategy acceptable within a setting where the threshold is less than $100,000/QALY. These findings diverge from the FDA’s approval of pembrolizumab, for instance, in patients with recurrent metastatic cervical cancer and platinum failure, but limited to those whose tumours exhibit biomarker test positivity [80]. To shed light on these results, we can analyse the outcomes of the clinical trials KEYNOTE-826 [81], designed to assess progression-free survival (PFS) and overall survival (OS) in patients with cervical cancer following genetic testing, and EMPOWER [82], which focused on evaluating OS. While both trials achieved their primary endpoints, subgroup analyses raised discussions on the benefits of ICIs for all cervical cancer patients [83]. Despite the limitations of PD-L1 expression as a predictive biomarker, it still informs clinical decision-making and can contribute to pushing advances in immunotherapy research [84]. Therefore, it can be concluded that a combination of biomarkers should be employed to identify patients who derive the greatest benefit [83]. 

### Strengths and Limitations

This study has several strengths including a comprehensive literature search and the pioneering nature of the objective, as the economic impact and sustainability of health systems regarding immunotherapy and biomarker testing have not been thoroughly investigated yet. Furthermore, the review was executed using a systematic approach, providing a comprehensive overview of the cost-effectiveness of biomarker testing worldwide and over a wide range of tumours. 

Nevertheless, this review did have certain limitations. The primary limitation is related to the inclusion of studies involving prior PD-1 testing constituting only 3% (*n* = 2) of the overall analysis. These specific studies are confined to pembrolizumab treatment in NSCLC. Due to this limited representation and the statistical insignificance of PD-1 in a broader context, we faced challenges in conducting a separate analysis based on PD-1 versus PD-L1 biomarkers. Another limitation is inherently associated with the methodology employed for conducting cost-effectiveness analyses (CEAs). Indeed, no study can fully encompass all of the potential cost-related factors or account for uncertainties surrounding the factors under investigation such as global economic and market forces, variations in practice and referral patterns, or reimbursements specific to individual insurance companies. Additionally, the set willingness-to-pay (WTP) thresholds do not incorporate other tax-related factors such as the financial impact on patients and supportive care providers, the utilisation of other costly cancer therapies, or indirect cost components such as the ability to return to work and contribute to the workforce and economy [3].

## 5. Conclusions

This systematic review highlights that PD-1 and PD-L1 overexpression, when used in combination with immune checkpoint inhibitor (ICI) therapy, could represent a cost-effective strategy for treating NSCLC as a first-line treatment with pembrolizumab and with both first- and second-line nivolumab, but also for renal cell and urothelial carcinoma. However, the cost-effectiveness is diminished when the willingness-to-pay (WTP) threshold falls below $100,000/QALY. Therefore, the economic impact of biomarkers upstream of the choice of the specific therapy represents an imperative to validate its effectiveness, the eventual relationship with the quality of life, and economic sustainability. A biomarker testing approach is therefore a virtuous model to invest in, providing the patient with a greater chance of receiving increasingly effective therapy and minimising adverse events due to the administration of untargeted therapies, resulting in an undoubted improvement in quality of life, while also optimising the management of health care resources.

## Figures and Tables

**Figure 1 cancers-16-00995-f001:**
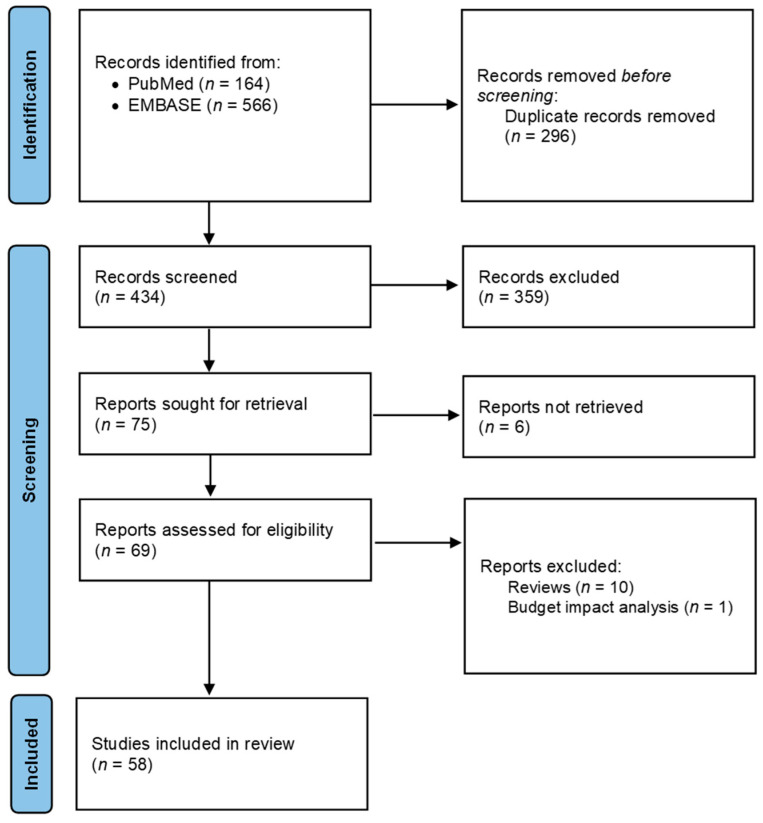
PRISMA diagram of the review’s systematic searches.

**Figure 2 cancers-16-00995-f002:**
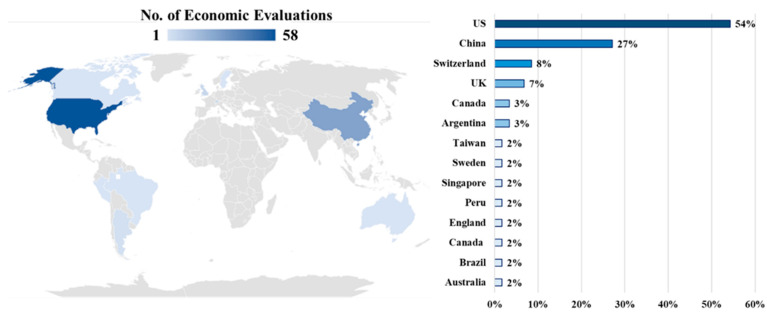
Global distribution of economic evaluations of an ICI associated with a biomarker test.

**Figure 3 cancers-16-00995-f003:**
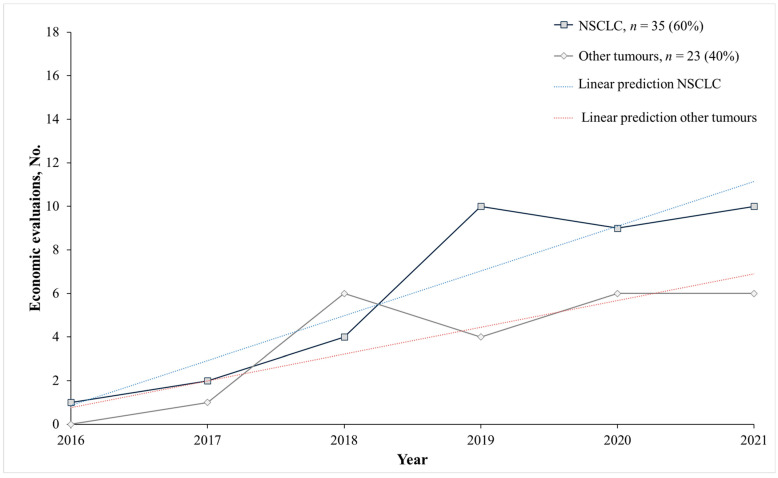
Economic evaluations of an ICI associated with a biomarker test per year focused on solid tumour diagnoses. Other tumours: Urothelial, bladded and renal cell cancer, cervical cancer, gastric cancer, head and neck cancer squamous cell carcinoma; melanoma; Merkel cell carcinoma, triple-negative breast cancer. Abbreviations: Non-small cell lung cancer (NSCLC). Notes: Dotted lines are estimates of linear predictions on the yearly trend of economic evaluations of ICIs associated with biomarker testing in NSCLC (blue dotted line) and other solid tumours (red dotted line).

**Figure 4 cancers-16-00995-f004:**
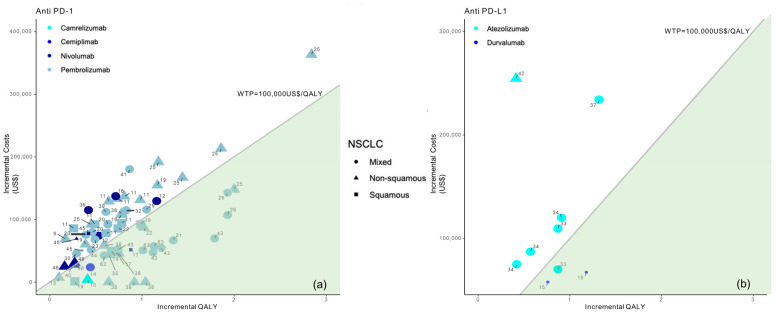
Results from the different studies on a cost-effectiveness plane for NSCLC patients according to anti PD-1 (**a**) or anti PD-L1 therapy (**b**). Note: Studies in the figure are identified by reference number.

**Figure 5 cancers-16-00995-f005:**
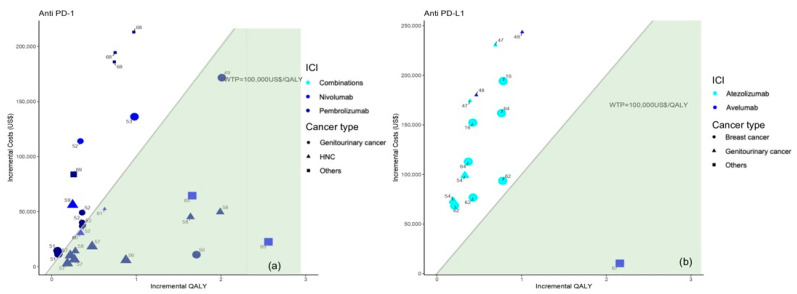
Results from the different studies on a cost-effectiveness plane for patients with cancer other than NSCLC according to anti PD-1 (**a**) or anti PD-L1 therapy (**b**). Note: Studies in the figure are identified by reference number.

**Table 1 cancers-16-00995-t001:** Characteristics of studies focusing on non-small cell lung cancer (NSCLC).

Study, Country, Year	ICI and Treatment Line	Comparators	Biomarker Test	Threshold (WTP)	Results	Cost-Effective
**PD-1 Target Therapy**						
**Barbier MC et al. Switzerland 2021 [9]**	Pembrolizumab 1L/2L	Pembrolizumab + SoC (chemotherapy)	PD-1	CHF 100,000/QALY	*Pembrolizumab plus SoC vs. pembrolizumab alone:* ICER CHF 475,299/QALY. *Pembrolizumab vs. SoC:* ICER of CHF 68,580/QALY.	yes
**Qiao L et al. China 2021 [10]**	Pembrolizumab 1L	SoC (chemotherapy)	PD-1	$150,000/QALY	*Pembrolizumab plus pemetrexed and platinum:* ICER $65,563/QALY.	yes
**PD-L1 Target Therapy**						
**Insinga RP et al. US 2021 [11]**	Pembrolizumab 1L	SoC (chemotherapy)	PD-L1, ALK, EGFR	$195,000/QALY	*Pembrolizumab plus SoC vs. SoC alone:* (a) Overall population, ICER $158,030/QALY; (b) Non-squamous and squamous patients, ICER $178,387/QALY.	yes
**Hu H et al. China 2021 [12]**	Nivolumab plus ipilimumab 1L	SoC (chemotherapy)	PD-L1	$100,000–$150,000/QALY	*Nivolumab plus ipilimumab*: (a) PD-L1 TPS ≥ 50%, ICER $107,403.72/QALY; (b) PD-L1 TPS ≥ 1%, $133,732.20/QALY; (c) PD-L1 TPS < 1%, $172,589.15/QALY.	yes
**Liu Q et al. US 2021 [13]**	Cemiplimab; Pembrolizumab 1L	SoC (chemotherapy)	PD-L1, ALK, EGFR	$100,000/QALY	*Cemiplimab vs. pembrolizumab:* ICER $52,998/QALY. *Cemiplimab vs. atezolizumab:* Gain of 0.13 QALYs and a decreased cost of $104,642, resulting in its dominance of atezolizumab. *Pembrolizumab plus chemotherapy vs.* *(a) Cemiplimab*: ICER $393,359/QALY; *(b) Pembrolizumab*: ICER $190,994/QALY; *(c) Atezolizumab*: ICER $33,230/QALY.	yes
**Rothwell B et al. England 2021 [14]**	Nivolumab NS	SoC (chemotherapy)	PD-L1	£50,000/QALY	*Nivolumab vs. docetaxel:* (a) Squamous NSCLC: ICER £35,657/QALY; (b) Non-squamous NSCLC: ICER £38,703/QALY. *Analysis were conducted with a confidential NHS England (NHSE) PAS (Patient access scheme) discount specific to the CDF (Cancer Drugs Fund):* (a) Squamous NSCLC, £68,576/QALY; (b) Non-squamous NSCLC, £73,189/QALY.	yes
**Cheng S et al. US and China 2021 [15]**	Atezolizumab 1L	SoC (chemotherapy)	PD-L1	US: $100,000–$150,000/QALY China: $33,210/QALY	*US*, *atezolizumab vs. SoC:* ICER $123,424/QALY. *China*, *atezolizumab vs. SoC:* ICER $78,936/QALY.	US, yes; China, no
**Liu G et al. China 2021 [16]**	Atezolizumab 1L	SoC (chemotherapy)	PD-L1	$30,828/QALY	*Atezolizumab vs. SoC*: (a) High PD-L1, ICER $123,778.60/QALY; (b) High or intermediate PD-L1 TPS, $142,827.19/QALY; (c) Any PD-L1 TPS, $168,902.66/QALY.	no
**Cai Y et al. China 2021 [17]**	Pembrolizumab 1L	SoC (chemotherapy)	PD-L1	$33,581.22/QALY	*Pembrolizumab vs. SoC*: ICER $65,272/QALY	no
**Yang SC et al. US 2021 [18]**	Nivolumab plus ipilimumab 1L	Nivolumab plus ipilimumab + SoC (chemotherapy)	PD-L1	$150,000/QALY	*Nivolumab plus ipilimumab* vs. *chemotherapy*: ICER $239,072/QALY. *Nivolumab plus ipilimumab plus SoC vs. Nivolumab plus ipilimumab:* ICER $838,198/QALY.	no
**Peng Y et al. US 2021 [19]**	Atezolizumab 1L	SoC (chemotherapy)	PD-L1	$100,000–$150,000/QALY	*Atezolizumab vs. SoC:* ICER $170,730/QALY and ICER $108,205/LY	no
**Panje CM et al. Switzerland 2020 [20]**	Durvalumab 1L/2L	Placebo	PD-L1	CHF 100,000/QALY	*Durvalumab:* (a) Unselected PD-L1 TPS, ICER CHF 88,703/QALY; (b) PD-L1 TPS ≥ 1%, ICER CHF 66,131/QALY.	yes
**Weng X et al. US 2020 [21]**	Pembrolizumab 1L	SoC (chemotherapy)	PD-L1	$180,000/QALY	*Pembrolizumab vs. SoC:* (a) PD-L1 TPS of ≥50%, ICER $47,596/QALY; (b) PD-L1 TPS ≥ 20%, $47,184/QALY; (c) PD-L1 TPS ≥ 1%, and $68,061/QALY.	yes
**Li J et al. China 2020 [22]**	Nivolumab 1L	SoC (chemotherapy)	PD-L1	$150,000/QALY	*Nivolumab vs. SoC*: ICER $180,307/QALY and $115,528/LY.	yes
**Wu B et al. US and China 2020 [23]**	Pembrolizumab 1L	SoC (chemotherapy)	-PD-L1 TPS ≥ 50% and >1% -Without EGFR, ALK mutations	US: $150,000/QALY China: $29,196/QALY	US: *Pembrolizumab plus chemotherapy vs. SoC*, (a) Non-squamous NSCLC, ICER $122,248; (b) Squamous NSCLC, ICER $121,375/QALY. *Adding TPS50 or TPS1 pembrolizumab treatment in patients with PD-L1 TPS test:* (a) Non-squamous disease, PD-L1 TPS ≥ 50%, ICER $143,282/QALY; PD-L1 TPS ≥ 1%, ICER $127,661/QALY. (b) Squamous NSCLC, PD-L1 TPS ≥ 50%, ICER $131,495/QALY; PD-L1 TPS ≥ 1%, ICER $121,554/QALY. China: *Adding pembrolizumab:* ICER $40,000/QALY.	US, yes; China, no
**Loong HH et al. Hong Kong (China) 2020 [24]**	Pembrolizumab 1L	SoC (chemotherapy)	PD-L1 TPS ≥ 50%	HKD 1017,819/QALY ($130,490)	*Pembrolizumab with TPS ≥ 50% vs. platinum doublet chemotherapy:* ICER HKD 865,189 ($110,922)/QALY and HKD 697,462 ($89,419)/LY.	yes
**Wan N et al. US and China 2020 [25]**	Pembolizumab 1L	SoC (chemotherapy)	PD-L1	US: $100,000/QALY China: $27,351/QALY	*Pembolizumab vs. SoC:* (a) US, ICER $132 392/QALY (b) China, $92,533/QALY. *PD-L1 ≥ 1% base case:* (a) US, ICER $77,754/QALY; (b) China, ICER $56,768/QALY. *PD-L1 ≥ 50% base case:* (a) US, ICER $44,731/QALY; (b) China, ICER $34,388/QALY.	no
**Criss SD et al.** **US 2020 [26]**	Pembrolizumab 1L	SoC (chemotherapy)	PD-L1	$100,000/QALY	*Pembrolizumab combination therapy vs SoC:* (a) Base case model Charlson 0 *, ICER $173,919/QALY; (b) Base case model Charlson 1 *, ICER $175,165/QALY; (c) Base case model Charlson 2+ *, ICER $181,777/QALY (d) PD-L1-high model Charlson 0 *, ICER $147,406/QALY; (e) PD-L1-high model Charlson 1 *, $149,026/QALY; (f) PD-L1-high model Charlson 2+ *, $154,521/QALY.	no
**Liu Q et al. China 2020 [27]**	Nivolumab 2L	SoC (chemotherapy)	PD-L1	$63,564/QALY	*Base case analysis*, *nivolumab:* ICER $74,126/LY and ICUR $93,307/QALY *Subgroup analyses, nivolumab:* (a) Patients ≥ 65 years $85,171/QALY; (b) Female patients $85,273/QALY; (c) Patients with PD-L1 TPS at least 1% $90,309/QALY.	no
**Aziz MIA et al. Singapore (Asia) 2020 [28]**	Pembrolizumab 1L	SoC (chemotherapy)	PD-L1	SGD100,000/QALY	*Pembrolizumab vs. SoC:* ICER SGD167,692/QALY.	no
**She L et al. US 2019 [29]**	Pembrolizumab 1L	SoC (chemotherapy)	-PD-L1 TPS ≥ 50%, ≥20% and ≥1% -Without EGFR, ALK mutations	$150,000/QALY	*Pembrolizumab vs. SoC:* (a) PD-L1 TPS ≥ 50%, ICER $136,228.82/QALY; (b) PD-L1 TPS ≥ 20%, ICER $160,625.98 /QALY; (c) PD-L1 TPS ≥ 1%, ICER $179,530.17/QALY.	yes
**Bhadhuri A et al. Switzerland 2019 [30]**	Pembrolizumab 1L	SoC (chemotherapy)	PD-L1 TPS ≥ 50%	CHF 100,000/QALY	*Pembrolizumab vs. SoC*: CHF 57,402/QALY and CHF 45,531/LY.	yes
**Chouaid C et al. US 2019 [31]**	Pembrolizumab 1L	SoC (chemotherapy)	PD-L1 TPS ≥ 50%	€ 100,000€/QALY	*Pembrolizumab vs. SoC:* Squamous NSCLC, ICER €84,097/QALY and €66,825/LY. *Pembrolizumab vs. platinum-based chemotherapy (paclitaxel plus bevacizumab):* Non-squamous NSCLC, ICER €78,729/QALY and €62,846/LY.	yes
**Huang M et al. US 2019 [32]**	Pembrolizumab 1L	SoC (chemotherapy)	PD-L1 TPS ≥ 1%	$100,000–$150,000/QALY	*Pembrolizumab vs. SoC*: ICER $130,155/QALY and $106,617/LY.	yes
**Wan X et al. US 2019 [33]**	Atezolizumab, bevacizumab, carboplatin, and paclitaxel 1L	ICIs + SoC (bevacizumab, carboplatin, and paclitaxel (BCP)) and SoC (chemotherapy)	PD-L1	$100,000/QALY	*Atezolizumab, bevacizumab, carboplatin, and paclitaxel* (ABCP) vs. *bevacizumab, carboplatin, and paclitaxel* (BCP): Non-squamous NSCLC: ICER $568,967/QALY. ABCP vs. *carboplatin and paclitaxel* (CP): non-squamous NSCLC: ICER $516,114/QALY.	no
**Insinga RP et al. US 2019 [34]**	Pembrolizumab 1L	SoC (chemotherapy)	PD-L1 TPS ≥ 50% and 1–49%	$100,000/QALY	*Pembrolizumab plus SoC (carboplatin and paclitaxel or nab-paclitaxel) vs. SoC*: (a) All patients, ICER $86,293/QALY and $72,725/LY (b) PD-L1 TPS ≥ 50%, $99,777/QALY; (c) PD-L1 TPS 1–49%, $85,986/QALY; (d) PD-L1 TPS < 1, $87,507/QALY.	yes
**Zhou K et al. China 2019 [35]**	Pembrolizumab 1L	SoC (chemotherapy)	PD-L1 TPS ≥ 50%, ≥20% and ≥1%	$26,508/QALY (product per capita of China in 2018)	*Pembrolizumab monotherapy vs. SoC:* (a) PD-L1 TPS ≥ 50%: ICER $36,493/QALY; (b) PD-L1 TPS ≥ 20%: ICER $42,311/QALY; (c) PD-L1 TPS ≥ 1%: ICER $39,404/QALY.	no
**Liao W et al. China 2019 [36]**	Pembrolizumab 1L	SoC (chemotherapy)	PD-L1	$100,000/QALY	*Pembrolizumab vs. chemotherapy*: ICER $103,128/QALY	no
**Insinga RP et al. US 2018 [37]**	Pembrolizumab 1L	SoC (chemotherapy)	PD-L1 TPS ≥ 50%	$180,000/QALY	*Pembrolizumab plus SoC vs. SoC alone:* (a) All patients, ICER $104,823/QALY and $87,242/LY; (b) PD-L1 TPS ≥ 50%, ICER/QALY $103,402; (c) PD-L1 TPS 1–49%, ICER $66,837; (d) PD-L1 TPS < 1%, ICER $183,529. *Pembrolizumab plus SoC vs. pembrolizumab alone:* ICER $147,365/QALY.	yes
**Georgieva M et al. US, UK 2018 [38]**	Pembrolizumab 1L	SoC (chemotherapy)	PD-L1 TPS ≥ 50%	UK: $42,048/QALY US: $100,000/QALY	*Pembrolizumab vs. SoC:* (a) UK, ICER $81,000/QALY; (b) US, ICER $74,000/QALY.	US, yes; UK, no
**Aguiar PN Jr et al. US 2018 [39]**	Nivolumab; Atezolizumab; Pembrolizumab 2L	SoC (chemotherapy)	PD-L1	$100,000/QALY	*Nivolumab vs. docetaxel*: (a) Squamous tumours, ICER $155,605 and $91,034/LY; (b) Non-squamous tumours, ICER $187,685/QALY and $102,8965/LY. *Atezolizumab vs. docetaxel:* All histologies, ICER $215,802/QALY and $103,095/LY. *Pembrolizumab vs. docetaxel:* PD-L1 TPS ≥ 1, ICER $98,421/QALY and $49,007/LY.	yes
**Hu X et al., UK 2018 [40]**	Pembrolizumab 1L	SoC (chemotherapy)	PD-L1	£50,000/QALY	*Pembrolizumab vs. SoC in PD-L1 positive patients:* ICER £86,913/QALY.	no
**Aguiar P et al. Argentina, Brazil, Peru 2018 [41]**	Pembrolizumab 1L/2L	SoC (chemotherapy)	PD-L1	Three-times the GDP per capita of each country, according to the World Health Organisation’s cost-effective definition	*First-line treatment with pembrolizumab vs. SoC:* (a) Brazil, ICER $63,424/QALY; (b) Argentina, ICER $139,351/QALY; (c) Peru, ICER $45,866/QALY. *Second-line treatment with pembrolizumab vs. SoC:* (a) Brazil, ICER $168,115/QALY; (b) Argentina, ICER $223,971/QALY; (c) Peru, ICER $170,383/QALY.	no
**Huang M et al. US 2017 [42]**	Pembrolizumab 1L	SoC (chemotherapy)	PD-L1 TPS ≥ 50%	$100,000/QALY	*Pembrolizumab vs. SoC:* ICER $114,194/QALY and $91,658/LY.	yes
**Matter-Walstra K et al. Switzerland 2016 [43]**	Nivolumab 1L	SoC (chemotherapy)	PD-L1	CHF 100,000/QALY	*Nivolumab vs. SoC in all treated patients:* ICER CHF 177,478/QALY. *Nivolumab vs. SoC in patients with PD-L1–positive tumours*: ICER CHF 124,891/QALY.	no

* Charlson score was calculated based on comorbidities present in the year before cancer diagnosis using the Comorbidity SAS Macro provided by the National Cancer Institute. Comorbidity burden level was divided into three groups—Charlson score equal to 0 (Charlson 0), Charlson score equal to 1 (Charlson 1), and Charlson score equal to 2+ (Charlson 2+).

**Table 2 cancers-16-00995-t002:** Characteristics of studies stratified by other solid tumours.

Study, Country, Year	ICI and Treatment Line	Comparators	Biomarker Test	Threshold (WTP)	Results	Cost-Effective
**Urothelial, bladder cancer, and renal cell carcinoma (RCC)**						
Qin S et al. US 2021 [44]	Atezolizumab (urothelial cancer) 1L	Placebo, SoC (gemcitabine and cisplatin or carboplatin)	PD-L1	$150,000/QALY	*Atezolizumab vs. placebo:* ICER $434,31/QALY and $325,352/LY *PD-L1 TPS at least 5% on immune cells*: ICER $325,236/QALY	no
Peng Y et al. US 2021 [45]	Avelumab (urothelial cancer) 1L	BSC alone	PD-L1	$150,000/QALY	*Avelumab vs. BSC alone:* (a) All patients (with unknown PD-L1 status), ICER $102,365/QALY; (b) PD-L1-guided strategy, ICER $106,253/QALY. *PD-L1-guided strategy vs. BSC:* ICER $105,360/QALY *PD-L1-guided strategy vs. avelumab:* ICER $122,653/QALY	yes
Hale O et al. US 2021 [46]	Pembrolizumab (Urothelial Carcer) 1L	SoC (chemotherapy)	PD-L1	$100,000/QALY	*Pembrolizumab vs. carboplatin plus gemcitabine:* ICER $78,925/QALY	yes
Patterson K et al. Sweden 2019 [47]	Pembrolizumab (Urothelial Cancer) 1L	SoC (Carboplatin + gemcitabine and gemcitabine monotherapy)	PD-L1	€100,000/QALY	*Pembrolizumab vs. gemcitabine plus carboplatin:* ICER €53,055/QALY and 42,967.32/LY *Pembrolizumab vs. gemcitabine monotherapy:* ICER €54,415/QALY and 44,025.65/LY	yes
Reinhorn D et al. US 2019 [48]	Nivolumab plus ipilimumab (RCC) 1L	SoC (chemotherapy)	PD-L1	$150,000/QALY	*Nivolumab and ipilimumab vs. sunitinib*: ICER $125,739/QALY	yes
Criss SD et al. US 2019 [49]	Pembrolizumab (Urothelial cancer) 2L	SoC (chemotherapy)	PD-L1	$150,000/QALY	*Pembrolizumab with PD-L1 positive tumours at a ≥1% expression threshold vs. SoC*: ICER $122,933/QALY *Pembrolizumab vs. Pembrolizumab with PD-L1 TPS ≥ 1%:* ICER $197,383/QALY	no
Sarfaty M et al. US 2018 [50]	Nivolumab (RCC) 2L	SoC (chemotherapy)	PD-L1	$100,000–$150,000/QALY	*Nivolumab vs. everolimus:* ICER $146 532/QALY *Nivolumab vs. placebo*: ICER $226,197/QALY	yes
Parmar A et al. Canada 2020 [51]	Atezolizumab (Bladder cancer) 2L	SoC (chemotherapy)	PD-L1	$100,000/QALY	*Atezolizumab vs. chemotherapy:* ICER CAD 430,652/QALY and CAD 292,228/LY *Scenario analysis, patients with PD-L1 expression levels of 5% or greater*: lower ICER CAD 334,387/QALY *Scenario analysis of observed compared with expected benefits demonstrated a higher icer, with a shorter time horizon:* CAD 928,950/QALY	no
Sarfaty M et al. US, UK, Canada, and Australia 2018 [52]	Pembrolizumab (Bladder Cancer) 2L	SoC (chemotherapy)	PD-L1	US: $100,000–150,000/QALY UK: $25,000–65,000/QALY Canada: $16,000–80,000/QALY Australia: $32,000–60,000/QALY	*Pembrolizumab vs. chemotherapy*: (a) US, $122,557/QALY; (b) UK, $91,995/QALY; (c) Canada, $90,099/QALY; (d) Australia, $99,966/QALY.	US yes; UK, Canada; Australia, no
**Head and neck cancer squamous cell carcinoma (HNSCC)**						
Wurcel V et al. Argentina 2021 [53]	Pembrolizumab 1L	Cetuximab + SoC (platinum + 5-fluorouracil)	PD-L1	$1,676,122/QALY	*Pembrolizumab monotherapy, vs. TT (cetuximab) + SoC (platinum + 5-fluorouracil):* ICER AR $135,801/LY and AR $170,985/QALY	yes
Zhou K et al. China 2020 [54]	Pembrolizumab 1L	SoC (chemotherapy)	PD-L1	$100,000/QALY	*Pembrolizumab monotherapy vs. cetuximab plus chemotherapy:* (a) Overall population, ICER $14,995/QALY; (b) CPS ≥ 1, $22,779/QALY; (c) CPS ≥ 20, ICER $39,535/QALY. *Pembrolizumab plus chemotherapy vs. standard therapy:* (a) Overall population, ICER $43,230/QALY; (b) CPS ≥ 1 $36,157/QALY; (c) CPS ≥ 20, ICER $55,679/QALY.	yes
Liu M et al. US and China 2019 [55]	Pembrolizumab 2L	SoC (chemotherapy)	PD-L1	US: $100,000/QALY China: $63,564/QALY	*Pembrolizumab group vs. PD-L1 CPS treatment*: (a) China, $7892/QALY; (b) US, $11,900/QALY.	yes
Zargar M et al. Canada 2018 [56]	Nivolumab NS	SoC (chemotherapy)	PD-L1	$100,000/QALY	*Nivolumab vs. docetaxel:* ICER $144,744/QALY	no
Hirschmann A et al. Switzerland 2018 [57]	Nivolumab 2L	SoC (chemotherapy)	PD-L1	CHF 100,000/QALY	*Nivolumab vs. SoC:* ICER CHF 102,957/QALY	yes
Ward MC et al. US 2017 [58]	Nivolumab NS	SoC (chemotherapy)	PD-L1	$100,000/QALY	*Nivolumab vs. SoC*: ICER $140,672/QALY	no
**Triple-negative breast cancer (TNBC)**						
Liu X et al. China 2021 [59]	Atezolizumab NS	SoC (chemotherapy)	PD-L1	$31,316/QALY	*Atezolizumab plus nab-paclitaxel versus nab-paclitaxel:* (a) ITT, ICER $176,056/QALY; (b) PD-L1(+), ICER $118,146/QALY; (c) PD-L1(–), ICER $323,077/QALY.	no
Weng X et al. US and China 2020 [60]	Atezolizumab 1L	SoC (chemotherapy)	PD-L1	US: $150,000/QALY China: $29,383/QALY	*Atezolizumab in combination with nab-paclitaxel (AnP) vs. nab-paclitaxel alone:* (a) US ITT population, ICER $331,996.89/QALY and $242,461.27/LY; (b) US PD-L1-positive patients, $229,359.88/QALY and $169,847.95/LY; (c) China ITT population, ICER $106,339.26/QALY and 77,660.83/LY; (d) China PD-L1-positive patients, $72,971.88/QALY and $54,037.89/LY.	no
Wu B et al. US 2020 [61]	Atezolizumab 1L	SoC (chemotherapy)	PD-L1	$200,000/QALY	*Atezolizumab plus nab-paclitaxel vs. nab-paclitaxel:* (a) Overall population, ICER $281,448/QALY; (b) Patients with CPS ≥ 1, ICER $196,073/QALY gained. *Atezolizumab plus nab-paclitaxel guiding by PD-L1 expression testing*: ICER $183,508/QALY.	(a) no (b) yes
**Melanoma**						
Meng Y et al. UK 2018 [62]	Nivolumab; Ipilimumab NS	Ipilimumab, dabrafenib, vemurafenib + SoC (dacarbazine)	PD-L1, BRAF	£50,000/QALY	*BRAF mutation-negative patients:* *(a) Nivolumab vs. ipilimumab:* ICER £24,483/QALY; *(b) Ipilimumab vs. dacarbazine:* ICER £22,589/QALY. *BRAF mutation-positive patients:* *(a) Nivolumab vs. ipilimumab (the only non-dominated comparator),* ICER £17,362/QALY.	yes
Tarhini A et al. US 2018 [63]	All 1L	TT (anti-PD-1 initiating sequences)	PD-L1, CTLA-4	$150,000/QALY	Anti *PD-1 + anti-CTLA-4 followed by chemotherapy vs. anti-PD-1 initiating sequences:* ICER $30,955/QALY.	yes
Chang WC et al. Taiwan 2021 [64]	Avelumab NS	SoC (chemotherapy)	PD-L1	$53,333.33/QALY	*Avelumab vs. BSC:* ICER USD 44 885.06/QALY. *Avelumab vs. SoC:* ICER USD 42 993.06/ QALY.	yes
**Cervical cancer**						
Shi Y et al. US 2021 [65]	Pembrolizumab NS	Placebo	PD-L1	$150,000/QALY	Pembrolizumab versus placebo: (a) ITT patients, ICER $247,663/QALY; (b) PD-L1 CPS ≥ 1, ICER $253,322/QALY; (c) PD-L1 CPS ≥ 10, ICER $214,212/QALY.	no
**Gastric cancer**						
Lauren B et al. US 2020 [66]	Pembrolizumab 2L	No second-line treatment	PD-L1	$100,000/QALY	*Pembrolizumab for MSI-H patients and ramucirumab/paclitaxel for all other patients vs. paclitaxel:* ICER $1,074,620/QALY. *Paclitaxel monotherapy for all patients:* ICER $53,705/QALY.	no

Abbreviations: Standard of care (SoC); target therapy (TT); combined positive score (CPS); best supportive care (BSC); incremental cost-effectiveness ratio (ICER); quality-adjusted life-year (QALY); Life-years (LYs); United States (US); United Kingdom (UK); immune checkpoint inhibitor (ICI).

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
