# Peer review of "Cost-Effectiveness of Treatment Optimisation with Biomarkers for Immunotherapy in Solid Tumours: A Systematic Review"

_cancers, 2024, doi:10.3390/cancers16050995_

Round 1
Reviewer 1 Report
Comments and Suggestions for Authors
This manuscript is a well-designed cost-effectiveness analysis of immune therapies using anit-PD-1 or PD-L1 antibodies in multiple solid tumors guided mainly by PD-L1 testing. The authors have systematically collected data from the past decade (2010 – 2022). They conclude that PD-L1 testing is an effective measure and cost-effective strategy for anti-PD-1/PD-L1 therapy mainly in NSCLC (lung cancer), urothelial, and renal cell carcinomas.
Major critiques:
- This is a well-designed and well-performed study. The authors had already published a study protocol in year 2021, which detailed the methodology used for the current study. Data collection is thorough and systematic. Data selection and processing are scrutinized and unbiased.
- Figure 4 and 5, how are the data points linked to results in table 1 and 2? For example, for cost-effective and cost-not-effective result, is it possible to mark dots in different color/shape in Figure 4 and 5? Line 142, the text claims 63% of studies are cost-effective for NSCLC and this is not easily appreciated in Figure 4. Also, is this increase of percentage (63%) statistically significant (e.g., proportion test or Fisher exact test may be used). Similar questions are applied to Figure 5 and Table 2 conclusions.
- The study should separate anti-PD-1 from anti-PD-L1, even though these are targeting the same pathway, to see whether the results can be improved.
Minor critiques:
- Figure 4/5 resolution is too low. The labels are not viewable.
- Line 145, “PD-1 assessment” may not be accurate since most biomarker testing in Table 2 are PD-L1 not PD-1.
Author Response
1. Summary
Thank you very much for taking the time to review this manuscript. Please find the detailed responses below and the corresponding revisions/corrections in track changes in the re-submitted files.
2. Point-by-point response to Comments and Suggestions for Authors Comments 1: This manuscript is a well-designed cost-effectiveness analysis of immune therapies using anit-PD-1 or PD-L1 antibodies in multiple solid tumors guided mainly by PD-L1 testing. The authors have systematically collected data from the past decade (2010 – 2022). They conclude that PD-L1 testing is an effective measure and cost-effective strategy for anti-PD-1/PD-L1 therapy mainly in NSCLC (lung cancer), urothelial, and renal cell carcinomas.
Response 1: Dear reviewer, we sincerely appreciate your insightful observation and valuable comments during the manuscript review. Your feedback has been instrumental in refining our work, and we are pleased to inform you that we have addressed all the constructive comments in the revised version that we are submitting.
Comments 2: Major critique 1: This is a well-designed and well-performed study. The authors had already published a study protocol in year 2021, which detailed the methodology used for the current study. Data collection is thorough and systematic. Data selection and processing are scrutinized and unbiased.
Response 2: Many thanks for your positive assessment regarding the study methodology.
Comments 3: Major critique 2: Figure 4 and 5, how are the data points linked to results in table 1 and 2? For example, for cost-effective and cost-not-effective result, is it possible to mark dots in different color/shape in Figure 4 and 5? Line 142, the text claims 63% of studies are cost-effective for NSCLC and this is not easily appreciated in Figure 4. Also, is this increase of percentage (63%) statistically significant (e.g., proportion test or Fisher exact test may be used). Similar questions are applied to Figure 5 and Table 2 conclusions.
Response 3: Dear reviewer, we sincerely appreciate your observation, in order to make it easier to identify the data points for cost-effective studies, we have revised both Figure 4 and Figure 5 by highlighting in green the area below the WTP threshold that identifies cost-effective results. To what concern the proportion of studies resulting to be cost-effective for both NSCLC and other tumors, in both cases the proportion resulted not significantly different from 50% and the p-value associated with the exact binomial test was not significant (p-value=0.176 for NSCLC and p=1 for other tumors). To do not put a lot of emphasis in those proportions the text was slightly modified.
Comments 4: Major critique 3: The study should separate anti-PD-1 from anti-PD-L1, even though these are targeting the same pathway, to see whether the results can be improved.
Response 4: Dear Reviewer, we greatly appreciate your observation, and we agree with your suggestion. In fact, this consideration was an aspect we had contemplated from the inception of the study. However, as reflected in Table 1 and Table 2, the study results demonstrate that,
2
in the overall analysis, 97% of the included studies (n=56) evaluated the cost-effectiveness ratio of an ICI with prior PD-L1 testing, while only 3% (n=2) incorporated prior PD-1 testing. It is noteworthy that these two studies encompassing prior PD-1 testing exclusively pertain to pembrolizumab treatment and are limited to NSCLC. Consequently, we faced constraints in splitting the analysis based on biomarker type. Nonetheless, we have taken steps to explicitly address this aspect in both the results and the study limitations. Please find these additions in the revised version of the manuscript.
Comments 5: Minor critique 1: Figure 4/5 resolution is too low. The labels are not viewable.
Response 5: We have revised and improved figures as suggested. The updated versions are submitted in a separate zip file.
Comments 6: Minor critique 2: Line 145, “PD-1 assessment” may not be accurate since most biomarker testing in Table 2 are PD-L1 not PD-1.
Response 6: We concur with your observation and appreciate your bringing it to our attention. Consequently, we have taken steps to rectify the statement as follows: “Particularly, pembrolizumab (n=17) [9,11,25–27,29,30,13,14,16,19–23] was cost-effective with respect to chemotherapy (SoC) when associated with a previous PD-1/ PD-L1 assessment, in first line treatment of NSCLC”.

Reviewer 2 Report
Comments and Suggestions for Authors
Sara Mucherino and co-authors present a high quality and well-written systematic review manuscript focused on cost-effectiveness of treatment optimisation with biomarkers for immunotherapy in solid tumours.
Authors aimed to provide a snapshot of the current state-of-the-art about the cost-effectiveness, cost-utility or net-monetary benefit for the use of predictive biomarkers in solid tumours treated with ICIs as tools for customizing immunotherapy, as part of a funded Italian National Research Project.
Authors investigated the health economic evaluations of predictive biomarker testing in solid tumors treated with immune checkpoint inhibitors. The focus was predominantly on non-small cell lung cancer (65%) and other solid tumors (40%). Among NSCLC studies, 21 out of 35 demonstrated cost-effectiveness, notably for pembrolizumab as first-line treatment when preceded by PD-L1 assessment, cost-effective at a threshold of $100,000/QALY compared to Standard of Care. However, for bladder, cervical, and triple-negative breast cancers, no economic evaluations met the affordability threshold of $100,000/QALY.
Overall, the review found PD-L1 expression associated with ICI treatment to be a cost-effective strategy, particularly in NSCLC, urothelial, and renal cell carcinoma. The findings suggest the potential value of predictive biomarker testing, specifically with pembrolizumab in NSCLC, while indicating challenges in achieving cost-effectiveness for certain other solid tumors.
Finally, authors conclude that that PD-L1 overexpression, when used in combination with immune checkpoint inhibitor therapy, represents a cost-effective strategy for treating major solid tumors, particularly in the case of NSCLC as a first-line treatment with pembrolizumab and with both first- and second-line nivolumab, but also for renal cell and urothelial carcinoma. Altogether, the economic impact of biomarkers upstream of the choice of the specific therapy represents an imperative to validate its effectiveness, the eventual relationship with the quality of life and economic sustainability.
Overall, the manuscript is highly valuable for the scientific community and should be accepted for publication.
======================
Other comments to authors:
1) Please check for typos throughout the manuscript.
2) Please improve figures/tables where appropriate.
3) With regards to immunotherapy of solid tumors– authors are kindly encouraged to cite the following article that describes novel approach for targeting solid tumor with immunotherapy.
DOI: 10.3390/biomedicines11020626
Author Response
1. Summary
Thank you very much for taking the time to review this manuscript. Please find the detailed responses below and the corresponding revisions/corrections in track changes in the re-submitted files.
2. Point-by-point response to Comments and Suggestions for Authors Comments 1: Sara Mucherino and co-authors present a high quality and well-written systematic review manuscript focused on cost-effectiveness of treatment optimisation with biomarkers for immunotherapy in solid tumours. Authors aimed to provide a snapshot of the current state-of-the-art about the cost-effectiveness, cost-utility or net-monetary benefit for the use of predictive biomarkers in solid tumours treated with ICIs as tools for customizing immunotherapy, as part of a funded Italian National Research Project. Authors investigated the health economic evaluations of predictive biomarker testing in solid tumors treated with immune checkpoint inhibitors. The focus was predominantly on non-small cell lung cancer (65%) and other solid tumors (40%). Among NSCLC studies, 21 out of 35 demonstrated cost-effectiveness, notably for pembrolizumab as first-line treatment when preceded by PD-L1 assessment, cost-effective at a threshold of $100,000/QALY compared to Standard of Care. However, for bladder, cervical, and triple-negative breast cancers, no economic evaluations met the affordability threshold of $100,000/QALY. Overall, the review found PD-L1 expression associated with ICI treatment to be a cost-effective strategy, particularly in NSCLC, urothelial, and renal cell carcinoma. The findings suggest the potential value of predictive biomarker testing, specifically with pembrolizumab in NSCLC, while indicating challenges in achieving cost-effectiveness for certain other solid tumors. Finally, authors conclude that that PD-L1 overexpression, when used in combination with immune checkpoint inhibitor therapy, represents a cost-effective strategy for treating major solid tumors, particularly in the case of NSCLC as a first-line treatment with pembrolizumab and with both first- and second-line nivolumab, but also for renal cell and urothelial carcinoma. Altogether, the economic impact of biomarkers upstream of the choice of the specific therapy represents an imperative to validate its effectiveness, the eventual relationship with the quality of life and economic sustainability. Overall, the manuscript is highly valuable for the scientific community and should be accepted for publication.
Response 1: Dear Reviewer, many thanks for your thoughtful and comprehensive review of our manuscript. We sincerely appreciate your positive feedback and recognition of the high quality and relevance of our systematic review on the cost-effectiveness of treatment optimization with biomarkers for immunotherapy in solid tumors. Your detailed summary of our findings, particularly the emphasis on the cost-effectiveness of PD-L1 expression associated with ICI treatment in NSCLC and other major solid tumors, is invaluable. We are grateful for your encouragement to publish our work.
Comments 2: Please check for typos throughout the manuscript.
Response 2: Thank you for your comment. We have carefully reviewed the entire
2
manuscript and addressed the typos. The revised version is resubmitted with improvements highlighted using track changes.
Comments 3: Please improve figures/tables where appropriate.
Response 3: We have revised and improved figures as suggested. The updated versions are included in the resubmitted manuscript.
Comments 4: With regards to immunotherapy of solid tumors– authors are kindly encouraged to cite the following article that describes novel approach for targeting solid tumor with immunotherapy. DOI: 10.3390/biomedicines11020626
Response 4: Thank you for the insightful suggestion. We appreciate your recommendation to cite the above article. We have incorporated this reference into the revised version of the manuscript.

Round 2
Reviewer 1 Report
Comments and Suggestions for Authors
Previous Major critique 3: The study should separate anti-PD-1 from anti-PD-L1, even though these are targeting the same pathway, to see whether the results can be improved.
Author's Response 4: Dear Reviewer, ...... as reflected in Table 1 and Table 2, the study results demonstrate that, in the overall analysis, 97% of the included studies (n=56) evaluated the cost-effectiveness ratio of an ICI with prior PD-L1 testing, while only 3% (n=2) incorporated prior PD-1 testing. ....
NOTE: The reviewer is asking on stratification of anti-PD1 and anti-PDL1 therapy, regardless of biomarker testing.
Comments on the Quality of English LanguageEnglish is good
Author Response
Response to Reviewer 1 Comments Round 2
|
||
1. Summary |
|
|
Thank you very much for taking the time to review this manuscript. Please find the detailed responses below and the corresponding revisions/corrections in track changes in the re-submitted files.
|
||
2. Point-by-point response to Comments and Suggestions for Authors |
||
Comments 1: Previous Major critique 3: The study should separate anti-PD-1 from anti-PD-L1, even though these are targeting the same pathway, to see whether the results can be improved. Author's Response 4: Dear Reviewer, ...... as reflected in Table 1 and Table 2, the study results demonstrate that, in the overall analysis, 97% of the included studies (n=56) evaluated the cost-effectiveness ratio of an ICI with prior PD-L1 testing, while only 3% (n=2) incorporated prior PD-1 testing. .... NOTE: The reviewer is asking on stratification of anti-PD1 and anti-PDL1 therapy, regardless of biomarker testing. |
||
Response 1: Dear reviewer, as previously mentioned, we understand your request and addressed now your request. In this newly submitted version, we have separated the results arising from PD-1 target therapy and PD-L1 target therapy (please refer specifically to Table 1, revised). Table 1 now shows that cost-effectiveness studies conducted on PD-1 targeted therapies (specifically pembrolizumab prescribed as a first-line treatment for NSCLC) are cost-effective (Swedish study [ref 9] and Chinese study [ref 13], both from 2021). Regarding PD-L1 targeted therapy, however, different results are observed depending on the type of pharmacological treatment and the type of solid tumor diagnosis, with a majority of studies confirming, in this case as well, that these therapies are cost-effective when WTP falls below $100,000/QALY.
. |

Round 3
Reviewer 1 Report
Comments and Suggestions for Authors
The study included mainly anti-PD1 therapy (nivolumab, pembrolizumab, cemiplimab, durvalumab) but also included some anti-PD-L1 therapy (atezolizumab, avelumab) studies. As shown in original and revised figure 4, cases under the WTP line and above the WTP line are roughly the same. This does not support the conclusion that PD-L1 test associated with ICI is cost-effective. Please remove anti-PD-L1 studies (atezolizumab, avelumab) and check whether this can improve the results. If not, the authors should consider to tune down their claim of ICI as cost-effective to cost-neutral.
Author Response
1. Summary |
|
|
Thank you very much for taking the time to review this manuscript. Please find the detailed responses below and the corresponding revisions/corrections in track changes in the re-submitted files.
|
||
2. Point-by-point response to Comments and Suggestions for Authors |
||
Comments 1: The study included mainly anti-PD1 therapy (nivolumab, pembrolizumab, cemiplimab, durvalumab) but also included some anti-PD-L1 therapy (atezolizumab, avelumab) studies. As shown in original and revised figure 4, cases under the WTP line and above the WTP line are roughly the same. This does not support the conclusion that PD-L1 test associated with ICI is cost-effective. Please remove anti-PD-L1 studies (atezolizumab, avelumab) and check whether this can improve the results. If not, the authors should consider to tune down their claim of ICI as cost-effective to cost-neutral. |
||
Response 1: Dear reviewer, we sincerely appreciate the substantial suggestion you have provided for our work. Indeed, we took some additional time to reanalyze the data separately for PD-1 and PD-L1, as per your recommendation. In this revised version that we are submitting, you will find updated results, including separate cost-effectiveness planes for PD-1 and PD-L1, reflected in the new Figures 4 and 5. Furthermore, we have adapted these new findings into the results, discussions, and conclusions sections of the paper. Thank you very much for your contribution to the work, and we hope you are pleased with the outcome.
. |
